# Genetic Variants in the *TBC1D2B* Gene Are Associated with Ramon Syndrome and Hereditary Gingival Fibromatosis

**DOI:** 10.3390/ijms25168867

**Published:** 2024-08-15

**Authors:** Thatphicha Kularbkaew, Tipaporn Thongmak, Phan Sandeth, Teerada Daroontum, Callum S. Durward, Pichai Vittayakittipong, Paul Duke, Anak Iamaroon, Sompid Kintarak, Worrachet Intachai, Chumpol Ngamphiw, Sissades Tongsima, Peeranat Jatooratthawichot, Timothy C. Cox, James R. Ketudat Cairns, Piranit Kantaputra

**Affiliations:** 1Center of Excellence in Medical Genetics Research, Faculty of Dentistry, Chiang Mai University, Chiang Mai 50200, Thailand; ktonmaii@hotmail.com (T.K.); worrachet.intachai@gmail.com (W.I.); 2Division of Pediatric Dentistry, Faculty of Dentistry, Chiang Mai University, Chiang Mai 50200, Thailand; 3Pediatric Division, Hatyai Hospital, Songkhla 90110, Thailand; tipaing@gmail.com; 4Department of Oral and Maxillofacial Surgery, Preah Ang Duong Hospital, Phnom Penh 120201, Cambodia; phan_sandeth@yahoo.fr; 5Department of Pathology, Faculty of Medicine, Chiang Mai University, Chiang Mai 50200, Thailand; mewteerada@gmail.com; 6Faculty of Dentistry, University of Puthisastra, Phnom Penh 120201, Cambodia; callumspencerdurward@gmail.com; 7Department of Oral and Maxillofacial Surgery, Faculty of Dentistry, Prince of Songkla University, Songkhla 90110, Thailand; pichai.v@hotmail.com; 8Royal Adelaide Hospital, Adelaide, SA 5000, Australia; paulduke01@gmail.com; 9Department of Oral Biology and Diagnostic Sciences, Faculty of Dentistry, Chiang Mai University, Chiang Mai 50200, Thailand; iamaroon@yahoo.com; 10Department of Oral Diagnostic Sciences, Faculty of Dentistry, Prince of Songkla University, Songkhla 90110, Thailand; sompid.k@psu.ac.th; 11National Biobank of Thailand, National Center for Genetic Engineering and Biotechnology, Thailand Science Park, Pathum Thani 12120, Thailand; chumpol.nga@gmail.com (C.N.); sissades.ton@biotec.or.th (S.T.); 12School of Chemistry, Institute of Science, and Center for Biomolecular Structure, Function and Application, Suranaree University of Technology, Nakhon Ratchasima 30000, Thailand; pj.cbsfa@gmail.com (P.J.); cairns@sut.ac.th (J.R.K.C.); 13Departments of Oral & Craniofacial Sciences, School of Dentistry, and Pediatrics, School of Medicine, University of Missouri-Kansas City, Kansas City, MO 64108, USA; coxtc@umkc.edu

**Keywords:** fibromatosis, gingival, cherubism, epilepsy, whole exome sequencing, TBC1D2B, KREMEN2

## Abstract

Ramon syndrome (MIM 266270) is an extremely rare genetic syndrome, characterized by gingival fibromatosis, cherubism-like lesions, epilepsy, intellectual disability, hypertrichosis, short stature, juvenile rheumatoid arthritis, and ocular abnormalities. Hereditary or non-syndromic gingival fibromatosis (HGF) is also rare and considered to represent a heterogeneous group of disorders characterized by benign, slowly progressive, non-inflammatory gingival overgrowth. To date, two genes, *ELMO2* and *TBC1D2B*, have been linked to Ramon syndrome. The objective of this study was to further investigate the genetic variants associated with Ramon syndrome as well as HGF. Clinical, radiographic, histological, and immunohistochemical examinations were performed on affected individuals. Exome sequencing identified rare variants in *TBC1D2B* in both conditions: a novel homozygous variant (c.1879_1880del, p.Glu627LysfsTer61) in a Thai patient with Ramon syndrome and a rare heterozygous variant (c.2471A>G, p.Tyr824Cys) in a Cambodian family with HGF. A novel variant (c.892C>T, p.Arg298Cys) in *KREMEN2* was also identified in the individuals with HGF. With support from mutant protein modeling, our data suggest that *TBC1D2B* variants contribute to both Ramon syndrome and HGF, although variants in additional genes might also contribute to the pathogenesis of HGF.

## 1. Introduction

Ramon syndrome (MIM 266270) or neurodevelopmental disorder with seizures and gingival overgrowth (MIM 619323) is an extremely rare genetic syndrome that was first described in two Israeli siblings in 1967 by Dr. Yochanan Ramon [1]. It is associated with a variety of phenotypic manifestations that can significantly impact the quality of life of affected individuals. The clinical findings include cherubism-like lesions, gingival fibromatosis (gingival overgrowth or overgrown gingiva), epilepsy, intellectual disability, hypertrichosis, short stature, juvenile rheumatoid arthritis, and ocular abnormalities. The onset is in childhood [2,3,4]. The genetic analysis of the original patients was performed in 2018 and implicated a homozygous variant in *Engulfment and cell motility gene 2* (*ELMO2*; MIM 606421) as underlying Ramon syndrome [5]. However, a more recent study demonstrated that the loss of function of *Tbc1 domain family member 2B* (*TBC1D2B*; MIM 619152) was the cause of Ramon syndrome [6]. Three additional loss-of-function mutations in *TBC1D2B* were subsequently reported in separate individuals affected with a neurodevelopmental disorder, gingival overgrowth, cherubism-like lesions, seizures, and ocular abnormalities, consistent with a diagnosis of Ramon syndrome [7].

Gingival fibromatosis, which is one of the clinical features of Ramon syndrome, may be present as an isolated condition (hereditary or non-syndromic gingival fibromatosis) (HGF; MIM 135300, 605544, 609955, 611010) or in association with several syndromes (syndromic). In addition to Ramon syndrome, these include Zimmermann–Laband (MIM 135500), Rutherfurd (MIM 180900), Jones (MIM 135550), and Cowden (MIM 158350) syndromes [8,9,10,11]. In contrast, HGF is an extremely rare presentation which is characterized by benign, slowly progressive, non-inflammatory gingival overgrowth [12]. In most patients, overgrown gingiva begins at the time when the permanent teeth erupt; however, in rare occasions it may appear along with the eruption of the primary teeth [13,14]. Histologically, the gingival tissues of patients with HGF exhibit an increased number of collagen fiber bundles running in different directions with few fibroblasts [12]. Genetic heterogeneity of HGF is suggested by anomalies in syndromic presentations. To date, genetic variants in five genes, *SOS RAS/RAC guanine nucleotide exchange factor 1* (*SOS1*; MIM 182530), *RE1-silencing transcription factor* (*REST*; MIM 600571), *Zinc finger protein 862 (ZNF862*), *Anaplastic Lymphoma Kinase* (*ALK*; MIM 105590), and *CD36 Antigen* (*CD36*; MIM 173510) have been linked to HGF [15,16,17,18]. 

Gingival fibromatosis has been linked to abnormal activation of the epithelial-to-mesenchymal transition [19], which is an important process during normal development involving the downregulation of E-cadherin expression, subsequent loss of epithelial cell contact, and increased cell motility [20]. The treatment of cultured human gingival epithelial cells with transforming growth factor β-1 (TGFB1) has been shown to induce the loss of cell surface E-cadherin, the upregulation of the expression of Snail family transcriptional repressor 2 (SLUG), and the subsequent epithelial-to-mesenchymal transition [19,21], providing a model for the presentation of HGF.

The objective of this study was to further investigate the genetic variants associated with Ramon syndrome as well as HGF. Here, we provide further support for the role of *TBC1D2B* in Ramon syndrome with the identification a novel homozygous variant in a Thai patient. Importantly, we also report a rare heterozygous variant in the *TBC1D2B* gene together with a novel heterozygous variant in the *KREMEN2* gene in three family members with HGF, suggesting an overlap in the pathogenesis of HGF with Ramon syndrome. 

## 2. Results

### 2.1. Patient Reports

Patient 1, a 7-year-old Thai girl, had suffered from gingival fibromatosis affecting the maxilla and mandible since age four (Table 1; Figure 1a,b). A gingivectomy was performed to correct the overgrown gingiva, but recurrence was evident three years later. A histopathologic examination of the gingival tissue showed an elongation of the rete ridges in the stratified squamous epithelial layer. The underlying fibrous connective tissue was markedly thickened with the normal density of fibroblasts (hematoxylin-eosin, original magnification: ×20) (Figure 2a). A computerized tomography scan at age 7 years showed persistent open anterior fontanelle (Figure 1e) and cherubism-like lesions in the mandible (Figure 1e–j). The histological sections from the mandibular lesions demonstrated multinucleated osteoclast-like giant cells within a fibrous stroma (Figure 2b). In addition, the child started having seizure episodes at the age of 5 years and was diagnosed as having focal epilepsy. The antiseizure medication prescribed for her was phenytoin, which was subsequently changed to valproic acid. Epilepsy and left eye amblyopia were also noted, suggesting a diagnosis of Ramon syndrome, a syndromic form of gingival fibromatosis [1].

Patients 2–4 were of Cambodian ancestry (Table 1). Patient 2, a 92-year-old male, presented with HGF and cataracts (Figure 3a,d). His two sons, ages 64 (patient 3; Figure 3b,e) and 49 years (patient 4; Figure 3c,f), presented with HGF which had the onset at 5–6 years of age. A histopathological study of the gingiva of patient 3 revealed dysplastic changes of the epithelial cells characterized by increased mitoses, enlarged nuclei with prominent nucleoli, and an increased nuclear/cytoplasmic ratio, suggestive of epithelial dysplasia (Figure 4). A gingivectomy and the removal of all the permanent teeth were performed on patients 3 and 4 with no recurrence after 7 years. The panoramic radiographs of patients 3 and 4 showed dense mandibular bone with displacement of the teeth. Generalized alveolar bone loss was observed in both patients (Figure 3g,h).

### 2.2. Bioinformatic and Mutation Analysis

Exome sequencing was performed on all affected individuals. A bioinformatic analysis identified a homozygous single base deletion (c.1879_1880del) in the *TBC1D2B* gene in the patient with Ramon syndrome (patient 1) (Figure 5a). The novel variant is predicted to result in a frameshift (p.Glu627LysfsTer61) that would cause loss of the entire TCB/Rab-GAP domain (Figure 6a,c), although the location of the variant would also predict the mRNA to undergo nonsense-mediated decay. This variant is predicted to be disease causing by MutationTaster (Prob = 1) (https://www.mutationtaster.org; accessed on 28 September 2023) and damaging by SIFT (0.858) (https://bio.tools/sift; accessed on 28 September 2023).

An extremely rare heterozygous base substitution variant (c.2471A>G) in the *TBC1D2B* gene was identified in patients 2–4 with hereditary gingival fibromatosis (Figure 5b). The heterozygous missense variant (c.2471A>G; rs199928887) is located in the C-terminal TBC/Rab-GAP domain and represents a change from tyrosine at position 824 to a cysteine (Figure 6a,c). This variant has an allele frequency of 0.00001301 (gnomAD database, v4.1). The Combined Annotation Dependent Depletion (CADD) score of p.Tyr824Cys was 22.8 and it was predicted to be disease causing (0.999153978449563), probably damaging (0.999), and damaging (0.01) by MutationTaster, PolyPhen-2, and SIFT, respectively. Despite the location of the Tyr824Cys change, its heterozygosity raises the possibility that additional variants might also play a role in the presentation in family 2. In this regard, patients 2–4 also segregated for a novel heterozygous variant (c.892C>T; p.Arg298Cys) in the *Kringle domain-containing transmembrane protein 2* (*KREMEN2*; MIM 609899) (Figure 5c). The CADD score of p.Arg298Cys was 23.5 and it was predicted to be disease causing by MutationTaster (prob = 0.876965507429932) (https://www.mutationtaster.org; accessed on 28 September 2023) and probably damaging by PolyPhen-2 (0.994) (http://genetics.bwh.harvard.edu/pph2/ accessed on 28 September 2023).

### 2.3. Protein Models

Schematics of the TBC1D2B and KREMEN2 protein domain structures show the positions of the mutations relative to the major structural domains of the protein based on NCBI accession numbers NP_653173.1 and NP_757384.1, respectively (Figure 6a,b). The structural models for human TBC1D2B (UniProt: Q9UPU7) and KREMEN2 (UniProt: Q8NCW0) were acquired from the AlphaFold2 database at the European Bioinformatics Institute (https://alphafold.ebi.ac.uk; accessed on 28 September 2023). AlphaFold2 is a protein structural modeling system that utilizes artificial intelligence (AI) to produce highly accurate protein structural models in the absence of experimental structures [22] and the models generated had high reliability scores in the region of the mutations. A predicted 3-dimensional structure model of the TBC1D2B TBC/Rab-GAP domain (blue) in complex with Rab GTPase was generated by superposition of the TCB1D2B model with the crystal structure of the complex of the TCB1D2B homologue Gyp1p with Rab (PDB: 2G77, Pan, Eathiraj, Munson & Lambright, 2006) to show the position of Rab binding (Figure 6c). This entire domain is missing in the p.Glu627LysfsTer61 mutation, so no Rab binding is possible. The position of Tyr824 is shown at the beginning of α-helix 11, which makes part of the interface with the Rab protein. The interaction of Tyr824 with its neighbors helps position this helix for interaction. A cysteine at position 824 is too small to make most of these interactions, as shown in the expanded pictures. We therefore propose that the Tyr824Cys mutation disrupts or destabilizes the interaction with Rab to impact the activity of TBC1D2B.

A predicted 3-dimensional model of the KREMEN2 protein was also generated with the area around Arg298 expanded to show neighboring charged residues. Arg298Cys mutation in the CUB domain is predicted to cause a loss of charge on the surface (Figure 6d). The CUB domain of the KREMEN2 is important for protein-protein and glycosaminoglycan-protein interactions. This change in the CUB domain might disrupt its interaction with one or more partner proteins such as DKK1, a Wnt inhibitor protein.

### 2.4. Immunohistochemical Findings

An immunohistochemical analysis of patient 1 revealed that, in comparison to a normal gingival control, the expression of E-cadherin appeared to be decreased in the basal layer of the epidermis, while the expression of BCL-2 and β-catenin appeared to be increased in the basal layer and the suprabasal layer of the epidermis, respectively. In patient 1, there was an apparent reduction in β-catenin expression and an increase in BCL-2 expression in the lamina propria. There was no consistent difference in E-cadherin, BCL-2, and β-catenin expression between normal gingival tissue and the gingival tissue of patient 4 (Figure 7).

## 3. Discussion

### 3.1. Genetic Variant in the TBC1D2B Gene and Ramon Syndrome

Our finding of a novel homozygous truncating allele in patient 1 (c.1879_1880del; p.Glu627LysfsTer61) adds further support for the alteration in *TBC1D2B* as a major cause of Ramon syndrome, given that only eighteen cases of this syndrome have been reported since 1967 [1,2,3,4,5,6,23,24]. The syndrome is characterized by cherubism-like lesions, which are progressive cystic formations that can be found predominantly in the mandible, epilepsy, neurodevelopmental disorder, gingival fibromatosis, hypertrichosis, short stature, juvenile rheumatoid arthritis, and ocular abnormalities [1,2,3,4,5,24]. Ocular abnormalities include Axenfeld anomaly, retinopathy, pale anomalous optic disc, and visual loss [6,23]. The most consistent finding in patients with Ramon syndrome appeared to be gingival fibromatosis, which was found in all but one patient evaluated at eight months of age [6]. Bi-allelic variants resulting in loss of function of the ELMO2 [5] and TBC1D2B proteins [6] have been reported in Ramon syndrome. Recently, a novel bi-allelic frameshift variant in the *TBC1D2B* gene was also found in two siblings with gingival fibromatosis, fibrous dysplasia of face, mental deterioration, limb tremors, gait ataxia, and seizures [7].

Hereditary gingival fibromatosis is a rare condition of overgrown gingival tissue, which can be either isolated or syndromic, including Ramon syndrome, Zimmermann–Laband syndrome, gingival fibromatosis with hypertrichosis (MIM 135400), Robinow syndrome (MIM 180700), Jones syndrome, Rutherford syndrome, hyaline fibromatosis syndrome (MIM 228600), and Enamel-Renal-Gingival syndrome (MIM 204690). To date, genetic variants in five genes, *SOS1*, *REST*, *ZNF862*, *ALK*, and *CD36*, have been linked to hereditary gingival fibromatosis [15,16,17,18].

In addition to identifying a novel homozygous truncating allele in patient 1, the observation of a rare, heterozygous *TBC1D2B* missense variant that segregated with HGF in family 2 also suggests variants in this gene contribute to isolated gingival presentations.

### 3.2. Genetic Variants in the TBC1D2B Gene and Gingival Fibromatosis

#### 3.2.1. Genetic Mechanisms and Clinical Implications

*TBC1D2B* is known to regulate the activities of the small GTPases RAB5, RAB31, RAB7, and RAB22, which together play a role in endosomal trafficking. RAB5 and RAB7 are responsible for early and late endosomal trafficking, respectively [25]. Genetic variants in *TBC1D2B* have been shown to result in abnormal interaction with RAB31 [26] and abnormal activation of RAC1 [27], RAB5 [28], RAB7 [26], and RAB22 [29]. The abnormal activation of RABs, including RAB5, RAB7, and RAB22, leads to abnormal endosomal trafficking [25], increased internalization and degradation of E-cadherin [29], increased epithelial-to-mesenchymal transition [29], uncontrolled cell growth [30], and subsequent gingival fibromatosis [19] (Figure 8). Our immunohistochemical study showed decreased expression of E-cadherin in the gingiva of the patient with Ramon syndrome (patient 1), supporting the involvement of the epithelial-to-mesenchymal transition in the pathogenesis of Ramon syndrome. A pathogenic role for the epithelial-to-mesenchymal transition in gingival overgrowth is further supported by a study in rats showing decreased expression of E-cadherin in phenytoin-, nifedipine-, and cyclosporin A-induced gingival overgrowth tissue [19].

Aberrant activation of RAC1, via RAB5, leads to the dysregulation of actin cytoskeleton dynamics, organization, and function, which also affects TGFB signaling [31], as well as NFkB and NFACT1 signaling [32]. Dysregulation of TGFB signaling can also lead to gingival overgrowth through the increased production of IL-6, which leads to fibroblast overactivation and connective tissue accumulation [33] as well as decreased apoptotic activity through increased expression of CCN2, periostin, and members of the lysyl family of enzymes [34,35] (Figure 8).

#### 3.2.2. Immunohistochemistry Results and Clinical Implications

BCL-2 and β-catenin play important roles in inhibiting apoptosis and increasing Wnt/β-catenin signaling, respectively. The increased expression of BCL-2 and β-catenin in the gingival tissues of patient 1 suggested that the overgrown gingiva of the patient with Ramon syndrome was associated with increased inhibition of apoptosis and increased Wnt/β-catenin signaling. These findings are also supported by the overexpression of BCL-2 in the nifedipine- and phenytoin-induced gingiva overgrowth model [36,37] and the increased expression of β-catenin in the overgrown gingiva of rats treated with cyclosporine A [38]. In addition, the decreased expression of E-cadherin in the gingiva of patient 1 could also exacerbate the accumulation of free β-catenin, which then enters the nucleus and contributes to increased cell proliferation and changes in gingival morphology [39].

### 3.3. Genetic Variants in TBC1D2B and KREMEN2 and Hereditary Gingival Fibromatosis

#### 3.3.1. A Heterozygous Missense Variant in the *TBC1D2B* Gene in Patients 2–4

Patients 2–4 were found to carry a rare heterozygous missense variant (c.2471A>G) in the *TBC1D2B* gene. Of note, the heterozygous variants in *TBC1D2B* have never been reported to cause abnormalities in humans. The amino acid Tyr824 is highly conserved in the TCB1D2B protein (Figure 5d) and is located at the beginning of the eleventh α-helix in the TCB/Rab-GAP domain. Based on its homology to Gyp1p, another Rab-GAP, this α-helix directly interacts with RAB proteins [40], a finding supported by our AlphaFold model (Figure 6a,c).

#### 3.3.2. A Novel Heterozygous Variant in the *KREMEN2* Gene in Patients 2–4

In addition to the *TBC1D2B* variant, patients 2–4 also had a novel heterozygous variant (c.892C>T; p.Arg298Cys) in the *KREMEN2* gene. This variant is predicted to be disease causing and probably damaging by MutationTaster and PolyPhen-2, respectively. The CADD score of this variant was 23.5. The Arg to Cys change in the extracellular domain of KREMEN2 could disrupt its binding to partner proteins, either because of the change in amino acid charge or because the cysteine could disrupt normal disulfide bond formation within the protein (Figure 6b,d).

*KREMEN2* is an essential regulator of the Wnt/β-catenin signaling pathway that is involved in various embryonic developmental processes, bone formation, and tumorigenesis [41]. It can directly bind to LRP5/6 and block their interaction with DKK1/2 [42,43]. The aberrant activation of Wnt signaling by such a variant would be predicted to be associated with increased endocytosis of the Wnt ligand–receptor complex. In addition, studies showed that the increased expression of KREMEN2 appeared to promote tumorigenesis and metastasis in gastric cancer [44] and was associated with a worse prognosis in colon cancer [45].

We therefore posit that the genetic variant in *KREMEN2* might contribute to the HGF presentation in this family via exacerbating the already perturbed endosome trafficking as a result of the heterozygous *TBC1D2B* variant, leading to an increased epithelial-to-mesenchymal transition, uncontrolled cell growth, and subsequent gingival hyperplasia and fibromatosis. Indeed, it is increasingly appreciated that many human diseases are more accurately described genetically when two or more gene mutations are involved, a process known as digenic inheritance or oligogenic inheritance, respectively [46]. Thus, we propose here that the *TBC1D2B* variant acts as a primary driver and the *KREMEN2* variant as a genetic modifier.

#### 3.3.3. Genetic Variant in the *KREMEN2* Gene and Dense Mandibular Bone in Patients 3 and 4

In addition to gingival fibromatosis, patients 3 and 4 also had dense mandibular bone. Of note, *KREMEN2* is also critical for bone formation and *Kremen2*-KO mice have been demonstrated to have increased bone formation [41]. Furthermore, overexpression of *Kremen2* in mice resulted in severe osteoporosis with decreased osteoblast and increased osteoclast differentiation and function [41]. Therefore, the genetic variant in the *KREMEN2* gene may also contribute to the dense mandibular bone in patients 3 and 4.

### 3.4. Genetic Variant in the TBC1D2B Gene and Cherubism-like Lesions

As mentioned earlier, a cherubism-like lesion is one of the features commonly found in Ramon syndrome. A gain-of-function mutation in the *SH3 domain-binding protein 2* (*SH3BP2*; MIM 602104) on chromosome 4p16.3 has been reported to cause cherubism [47,48]. SH3BP2 cooperates with Vav proteins, which are guanine nucleotide exchange factors that activate the small GTPases Ras and Rac1 [49]. Deletion of Rac1 in osteoclasts has been reported to cause osteopetrosis; its deletion severely disrupts the cytoskeleton of osteoclasts [50]. As cherubism-like lesions are a part of Ramon syndrome, it is therefore expected that cherubism and Ramon syndrome share some pathogenesis pathways.

Patient 1 displayed cherubism-like lesions on both sides of the mandible, which was consistent with the other five patients carrying homozygous and compound heterozygous variants in the *TBC1D2B* gene reported in two previous studies [6,7]. Thus, the alteration in the TBC1D2B protein may also lead to a cherubism-like phenotype via the abnormal activation of downstream RAB5, overactivation of RAC1, abnormal NFkB, and NFACT1 signaling in osteoclast progenitor cells, which subsequently leads to increased osteoclast formation and hyperosteolysis [32,48,51] (Figure 8).

### 3.5. Genetic Variant in the TBC1D2B Gene and Seizures

Patient 1 had her first seizure at the age of five and was diagnosed with focal epilepsy. Seizures were also found in four patients with homozygous and compound heterozygous variants in the *TBC1D2B* gene reported in two previous studies but at different ages of onset: two at 19 years, one at 3 months, and one at 32 years of age [6,7]. It is proposed that aberrant TBC1D2B activity may lead to abnormal neural development and seizures via the overactivation of the RAC1 and RAF1/MAPK/ERK signaling pathways and dysregulation of TGFB signaling [52,53] (Figure 8).

### 3.6. Genetic Variants in the TBC1D2B Gene and Epithelial Dysplasia

The histopathological examination of the gingiva of patient 3 revealed dysplastic changes of the epithelial cells characterized by increased mitoses, enlarged nuclei with prominent nucleoli, and an increased nuclear/cytoplasmic ratio, which are suggestive of epithelial dysplasia. This presentation has never been documented in the previous studies of patients with *TBC1D2B* variants. Epithelial dysplasia has, however, been reported in a patient with HGF of unknown genetic etiology [54].

As mentioned above, TBC1D2B is known to regulate the activities of the small GTPases RAB5, RAB7, and RAB22, which together play a role in the epithelial-to-mesenchymal transition and E-cadherin internalization. Thus, it is hypothesized that aberrant TBC1D2B activity may lead to an increased epithelial-to-mesenchymal transition, degradation of E-cadherin, and subsequent epithelial dysplasia. In addition, the histopathological study of the gingiva of patient 1 showed elongation of the rete ridges in the stratified squamous epithelial layer (Figure 2a), which is a typical feature of gingival overgrowth. The elongated rete ridges observed in patient 1 were consistent with the previous studies and were suggested to be the result of increased epithelial plasticity and epithelial-to-mesenchymal transition [19,55] (Figure 8).

### 3.7. Genetic Variants in the TBC1D2B Gene and Recurrence of Overgrown Gingiva

Patient 1 underwent gingivectomy to correct the overgrowing gingiva at four years of age; recurrent gingival overgrowth was observed three years later. In contrast, a 7-year follow-up of patients 3 and 4 showed no evidence of the recurrence of gingival overgrowth after a gingivectomy and the removal of all teeth. It is known that the presence of teeth promotes the recurrence of gingival overgrowth after surgery. The overgrowth disappears or recedes when the corresponding teeth are extracted [56,57,58]. However, the exact role of teeth in promoting the recurrence of gingival overgrowth after surgery is unknown. What is known is that gingival fibroblasts from patients with recurrence show increased cell proliferation compared with fibroblasts from patients without recurrence. In addition, it has been noted that gingival overgrowth with dense connective tissue and abundant blood vessels and fibroblasts is more likely to recur after surgery [59]. A genetic predisposition may impact individual responses following surgical interventions and may interact with dental factors such as an oral microbiome. Consequently, variations in the genes associated with inflammation and tissue remodeling may result in differing susceptibilities to the recurrence of gingival overgrowth.

### 3.8. Future Research Directions

Future research and potential therapeutic strategies for hereditary gingival fibromatosis should emphasize the cellular processes involved, particularly the epithelial-mesenchymal transition and the Wnt signaling pathway. Additionally, it is important to investigate how these processes relate to other clinical manifestations of Ramon syndrome.

## 4. Materials and Methods

### 4.1. Ethics Statement

This study was conducted in accordance with the Declaration of Helsinki and national guidelines. Informed consent was obtained from the patients or the parents of the patients in accordance with regulations of the Human Experimentation Committee of the Faculty of Dentistry, Chiang Mai University (certificate of approval number 12/2023).

### 4.2. Patient Recruitment

Inclusion criteria for patient recruitment were patients diagnosed with gingival fibromatosis, regardless of age, gender, or whether other clinical features were present. The experimental period was from October 2022 to April 2024. A Thai patient with Ramon syndrome and her parents and three Cambodian patients from one family with HGF were enrolled in this study.

Patient 1 was a 7-year-old Thai girl with gingival fibromatosis in the maxilla and mandible, cherubism-like lesions in the mandible, a persistent opening of anterior fontanelle, epilepsy, and left eye amblyopia. Patients 2–4 were of Cambodian ancestry. Patient 2, a 92-year-old male, presented with HGF and cataracts. His two sons, ages 64 (patient 3) and 49 years (patient 4), presented with HGF.

### 4.3. Whole Exome Sequencing, Mutation Analysis, and Bioinformatic Analyses

The patients’ genomic DNA was isolated from blood samples in BD Vancutainer^®^ EDTA tubes (10.0 mL K_2_E, 18.0 mg EDTA) (BD-Plymouth, PL6 7BP, UK) for whole exome sequencing. Whole exome sequencing is a widely used technique that targets the majority of coding regions in the genome, where most disease-causing mutations are found. This method was performed on the Thai patient with Ramon syndrome and her two unaffected parents, as well as on the three Cambodian patients with hereditary gingival fibromatosis.

All prepared DNA samples were subjected to exome capture using the SureSelect V6+UTR-post kit (Agilent Technologies, Santa Clara, CA, USA). The average depth of the sequencing was 80× using the targeted capture SureSelect V6 kit (PR7000-0152; Agilent Technologies, Santa Clara, CA, USA), which also captures untranslated regions. We adopted GATK3.8 best practices to identify variants for each sample; the alignment of the raw sequencing FASTQ file with the human genome reference sequence, GRCh38+decoy, was performed using BWA-mem. The variant effect predictor (VEP) and the database of non-synonymous functional prediction (dbNSFP) were used to computationally assign effects to the resulting variants of each individual. The annotated variant calling format (VCF) files were stored in our in-house database that allows us to query pathogenic variants according to different modes of segregation. Furthermore, the variant allele frequencies were determined by comparing against public databases, including 1000 Genomes, gnomAD, GenomeAsia, and the recent Thai Reference Exome database. The prioritization of the variants was established based on multiple considerations.

Prioritized candidate variants of interest were validated and evaluated for segregation by PCR-based amplification and Sanger sequencing. The Sanger sequencing was done to verify the presence of variants in DNA samples. The DNA samples designated for sequencing were mixed with primers in tubes. The sequence primers used for *TBC1D2B* variants were as follows: Exon 9 for patient 1, forward: 5′-AAAGTACCACGTGTGCCTGAT-3′; reverse: 5′-CTGGATTCCGCCAGGAGAAG-3′. Exon 11 for patients 2–4, forward: 5′-GTCTGCTCGTCTGCTTACCC-3′; reverse: 5′-CCAGATCTAGTGCAGGCCG-3′. The sequence primers used for the *KREMEN2* variant for patients 2–4: forward: 5′-GTG TCC TGG TCC TCA GGA TG-3′; reverse: 5′-AGA GCG GGG TGG GAA CA-3′.

### 4.4. Structural Assessment of Variants

In order to understand possible effects of the mutations on the proteins that may explain the molecular basis of disease, molecular models of the proteins were obtained. The structural models for human TBC1D2B (UniProt: Q9UPU7) and KREMEN2 (UniProt: Q8NCW0) were acquired from the AlphaFold2 database at the European Bioinformatics Institute (https://alphafold.ebi.ac.uk). To investigate whether the mutations could affect interactions between TBC1D2B and Rab proteins, the predicted 3-dimensional structure model of the TBC1D2B TBC/Rab-GAP domain in complex with Rab GTPase was generated by superposition of this model with the crystal structure of the Gyp1p-Rab complex (PDB: 2G77, Pan, Eathiraj, Munson & Lambright, 2006) in the molecular viewing program PyMol v.2.6 (Schrödinger Inc., New York, NY, USA). All structures and mutations were visualized in PyMol v.2.6.

### 4.5. Immunohistochemistry

Formalin-fixed, paraffin-embedded gingival tissues were cut into 4–5 μm thick sections and placed on Superfrost Plus microscope slides. The slides were heated at 60 °C for one hour in an incubator to enhance tissue attachment and soften the paraffin. The sections were deparaffinized with xylene and rehydrated through graded ethanol into water. Antigen retrieval was performed by using CC1 (prediluted, pH 8.0) antigen retrieval solution (Ventana). The immunohistochemistry procedure was performed on a Ventana BenchMark ULTRA autostainer according to the standard protocol. The slides were incubated with primary antibodies. The following primary antibodies were used: E-cadherin, a major component of the epithelial adherens junction complex and a marker of the epithelial-mesenchymal transition process (clone NH-38, Dako, Santa Clara, USA, ready-to-use); BCL2, an inhibitor of apoptosis induced by programmed cell death stimuli (clone 124, Dako, USA, ready-to-use); and β-catenin, a key effector in the canonical Wnt signaling pathway and a crucial component of cadherin-based adherens junctions (clone β-catenin-1, code M3539; Dako, Santa Clara, USA, 3:100 dilution). The visualization process was performed with the Ultraview universal DAB IHC detection kit, and it was afterwards counterstained in hematoxylin and bluing solution. The slides were gently cleaned and dehydrated in graded ethanol and xylene. Finally, they were mounted with mounting media on a microscope slide.

## 5. Conclusions

TBC1D2B is a regulator of RAB5, RAB7, and RAB22, which are critical for endosome maturation and E-cadherin internalization [26,28,29]. Defective endosomal maturation and E-cadherin internalization as a result of the *TBC1D2B* mutations result in an increased epithelial-to-mesenchymal transition, uncontrolled cell growth, and subsequent gingival fibromatosis. We propose that the rare heterozygous variant in the *TBC1D2B* gene contributes to the pathogenesis of HGF, perhaps in concert with variants that disrupt either Wnt signaling, endosome maturation, or lysosomal activity. It appears that gingival fibromatosis in patients with *TBC1D2B* variants is a consequence of disrupted RAB-mediated endocytic trafficking as a result of the altered TBC1D2B protein. Future research and therapeutic strategies for hereditary gingival fibromatosis should focus on the cellular processes involved, particularly the epithelial-mesenchymal transition and the Wnt signaling pathway. Furthermore, examining the relationship between these processes and other clinical manifestations of Ramon syndrome is essential. It is crucial to perform a thorough examination and genetic testing in patients with HGF in order to provide an accurate diagnosis and genetic counseling in the future.

## Figures and Tables

**Figure 1 ijms-25-08867-f001:**
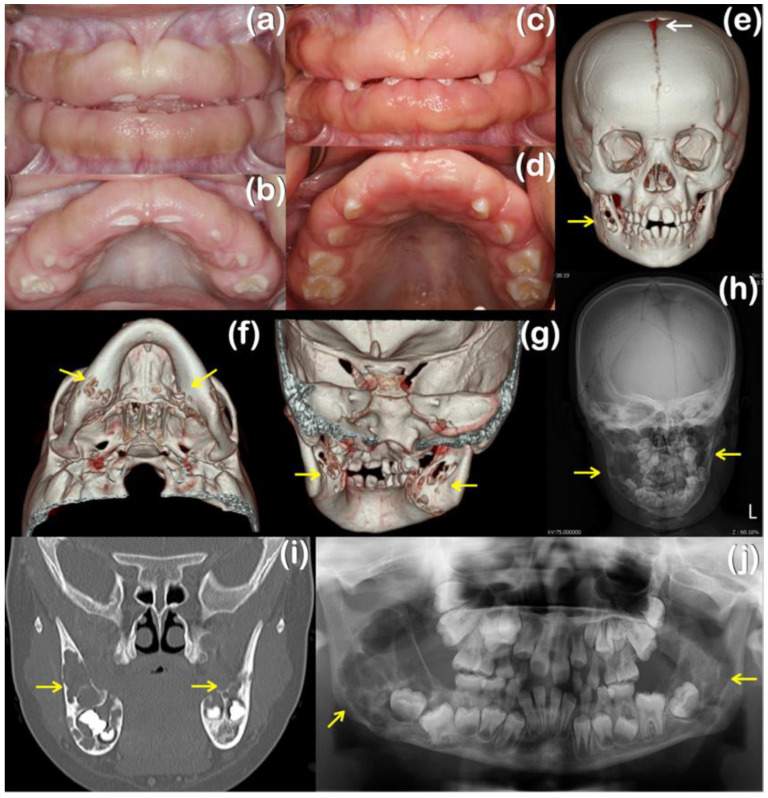
Patient 1. (**a**,**b**) Patient 1, aged 4 years, carrying a homozygous variant c.1879_1880del; p.Glu627LysfsTer61 in the *TBC1D2B* gene. Non-inflammatory gingival overgrowth is evident. (**c**,**d**) Patient 1, aged 7 years, demonstrating recurrent gingival overgrowth. (**e**–**i**) A computerized tomography scan showing expansile and perforated mandible (arrows) and a persistent opening of the anterior fontanelle (arrow). (**j**) Panoramic radiograph showing expansile mandible with cherubism-like lesions (arrows). Unerupted permanent mandibular molars are associated with cherubism-like lesions.

**Figure 2 ijms-25-08867-f002:**
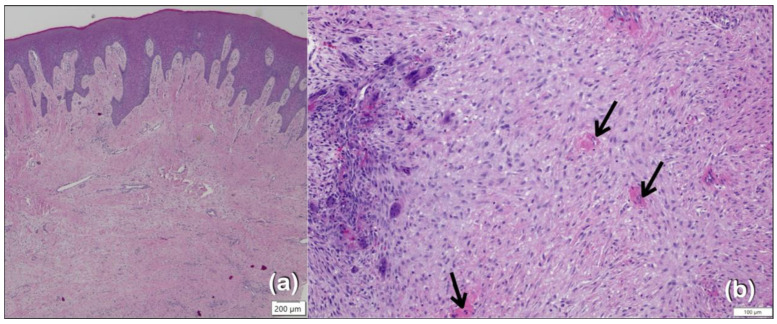
(**a**) Histopathologic examination of the gingival tissue of patient 1. The overlying surface stratified squamous epithelium demonstrates elongation of the rete ridges. The underlying fibrous connective tissue is markedly thickened with normal density of fibroblasts (hematoxylin-eosin, original magnification: ×20). (**b**) Histological section of the sample from of the mandible of patient 1 demonstrating multinucleated osteoclast-like giant cells (arrows) within a fibrous stroma, a characteristic feature of cherubism.

**Figure 3 ijms-25-08867-f003:**
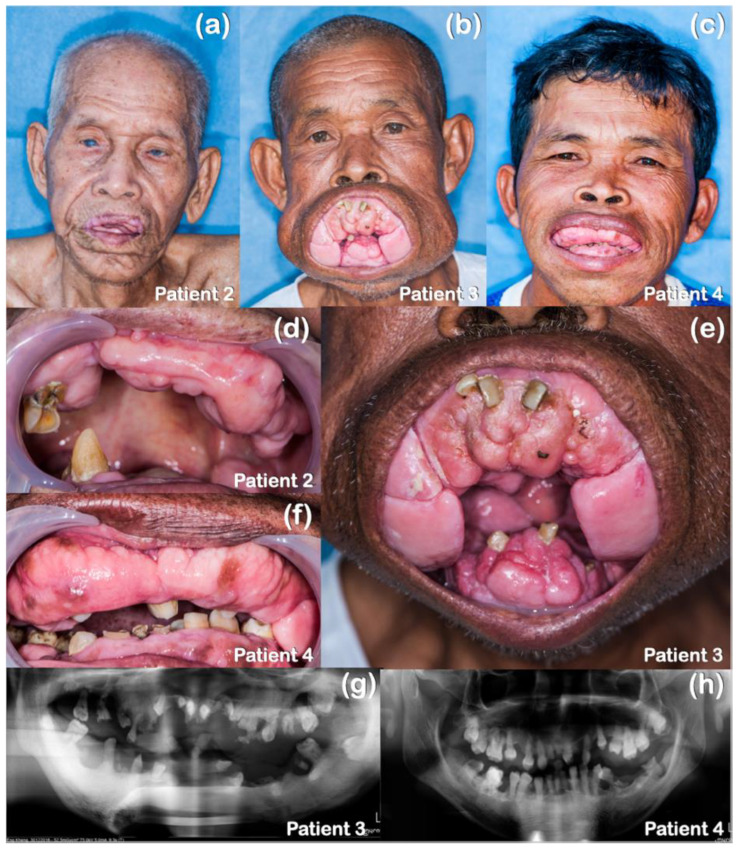
Patients 2–4 with HGF who carried a novel heterozygous variant in the *TBC1D2B* (p.Tyr824Cys) and *KREMEN2* (p.Arg259Cys) genes. (**a**) Patient 2 with cataracts and gingival overgrowth. (**b**) Patient 3 with severe gingival overgrowth. (**c**) Patient 4 with gingival overgrowth. (**d**) Gingival overgrowth in patient 2. (**e**) Severe gingival overgrowth in patient 3. (**f**) Gingival overgrowth in patient 4. (**g**) Panoramic radiograph of patient 3 showing dense mandibular bone, displacement of teeth, and generalized alveolar bone loss. (**h**) Panoramic radiograph of patient 4 showing dense mandibular bone of patient 4 with generalized alveolar bone loss.

**Figure 4 ijms-25-08867-f004:**
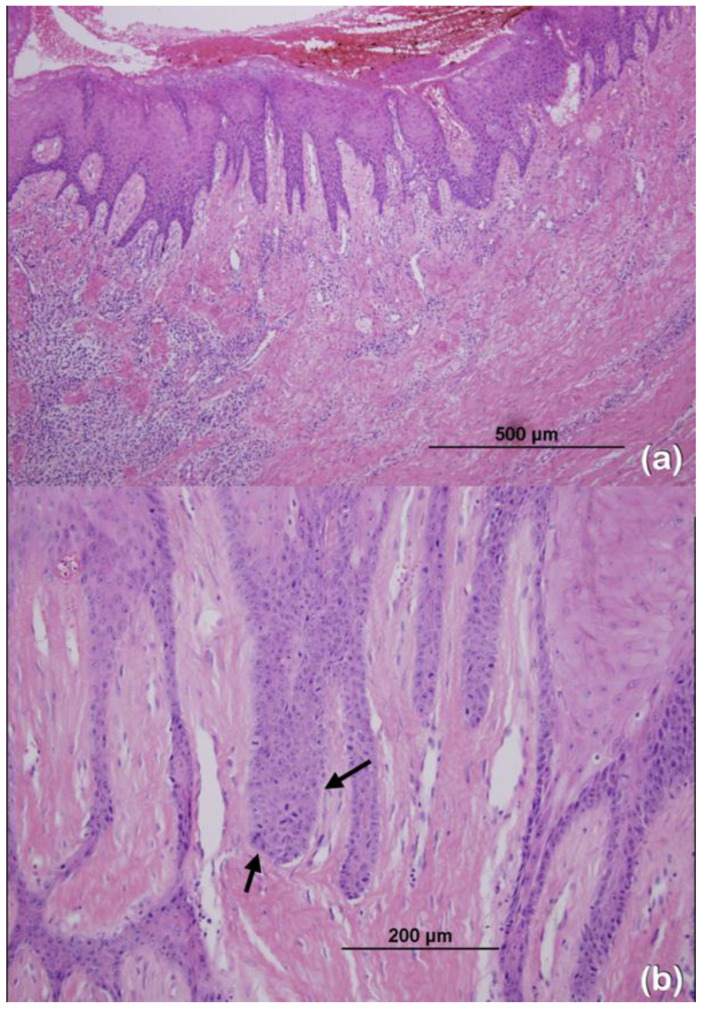
(**a**) Histopathologic examination of the gingival tissue of patient 3. The overlying surface stratified squamous epithelium demonstrates elongation of the rete ridges. The underlying fibrous connective tissue is markedly thickened with normal density of fibroblasts (hematoxylin-eosin, original magnification: ×20). (**b**) Dysplastic changes of epithelial cells characterized by increased mitoses, enlarged nuclei with prominent nucleoli (arrows), and increased nuclear/cytoplasmic ratios, suggestive of epithelial dysplasia.

**Figure 5 ijms-25-08867-f005:**
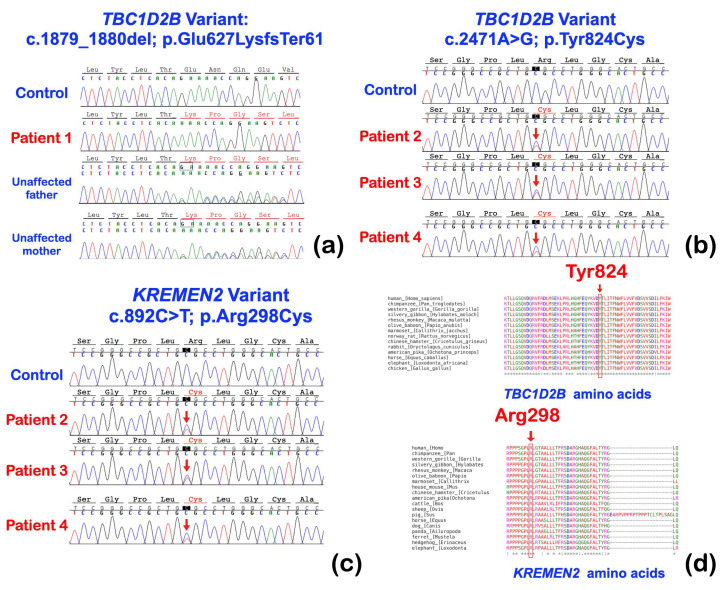
(**a**–**c**) Sequence chromatograms of *TBC1D2B* and *KREMEN2* variants in patients 1–4, controls, and their unaffected family members. (**d**) Conservation of amino acid residues. The amino acids Tyr824 and Arg298 are highly conserved across vertebrate species.

**Figure 6 ijms-25-08867-f006:**
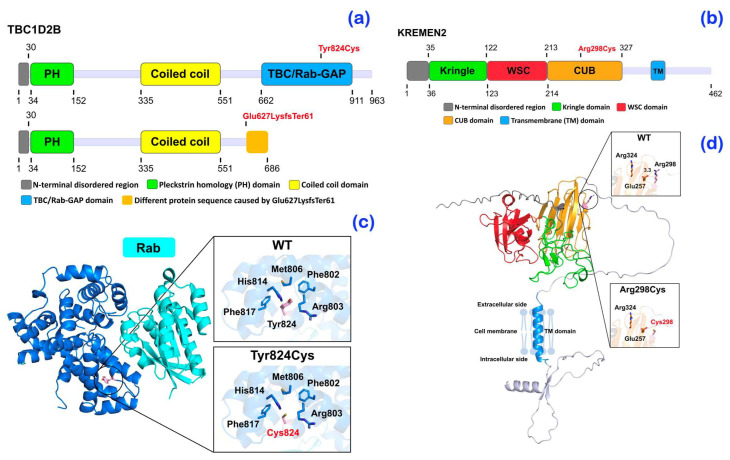
Protein changes and consequences. (**a**) Map of the TBC1D2B protein sequence. (**b**) Map of the KREMEN2 protein sequence. (**c**) Predicted 3-dimensional structure model of the TBC1D2B. (**d**) Predicted 3-dimensional model of KREMEN2. The protein domains in (**c**,**d**) are colored similar to their colors in the domain maps in (**a**,**b**). Side chains of mutated residues and their surrounding amino acid residues are shown in stick representation. Magenta carbon, red oxygen, blue nitrogen, and dark yellow sulfur atoms depict the mutations.

**Figure 7 ijms-25-08867-f007:**
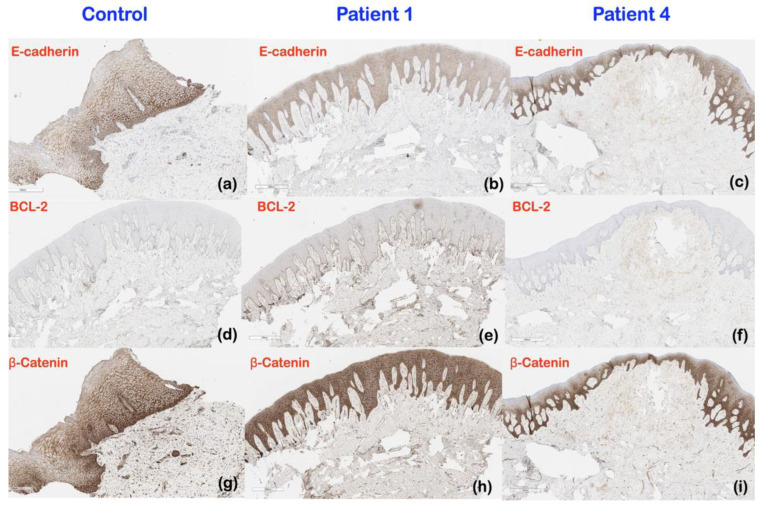
Representative images of immunohistochemistry study to determine E-cadherin, BCL-2, and β-catenin expression in (**a**,**d**,**g**) normal gingiva controls, (**b**,**e**,**h**) patient 1 (Ramon syndrome), and (**c**,**f**,**i**) patient 4 (HGF) (magnification, 8×). (**b**) E-cadherin showed decreased expression in the basal layer of the epidermis in patient 1. (**e**) BCL-2 showed increased expression in the basal layer of the epidermis and the lamina propria in patient 1. (**h**) β-catenin showed increased expression in the upper layer of the epidermis but decreased expression in the lamina propria of patient 1. No differences of (**a**,**c**) E-cadherin, (**d**,**f**) BCL-2, and (**g**,**i**) β-catenin expression were observed between the gingiva of patient 4 and the normal gingival tissue.

**Figure 8 ijms-25-08867-f008:**
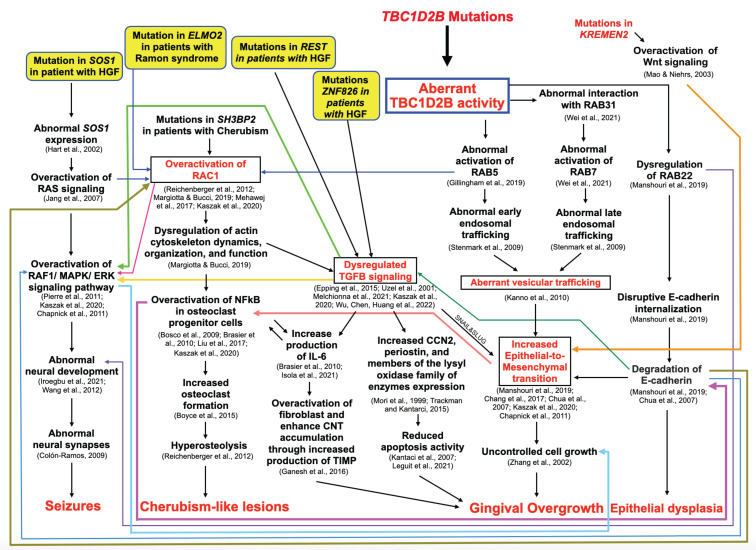
Flowchart showing hypothetical pathogenesis mechanisms leading to gingival fibromatosis, seizure, epithelial dysplasia, and cherubism-like lesions as the results of genetic variants in *TBC1D2B*, *SOS1*, *ELMO2*, and *REST*. The main pathogenetic mechanisms appear to be overactivation of RAC1, dysregulated TGFB signaling, and increased epithelial-to-mesenchymal transition. Refs. [1,2,3,4,5,6,7,8,9,10,11,12,13,14,15,16,17,18,19,20,21,23,24,25,26,27,28,29,30,31,32,33,34,35].

**Table 1 ijms-25-08867-t001:** Patients with the *TBC1D2B* and *KREMEN2* variants and their phenotypes.

Families	Patients	Phenotypes	Clinical Findings	Variants	Prediction/Ranking
1	1 (Female)	Ramon Syndrome	Gingival fibromatosis, cherubism-like lesions, a persistent open anterior fontanelle, focal epilepsy, left eye amblyopia	*TBC1D2B* variant NM_144572.2; NP_653173.1 c.1879_1880del p.Glu627LysfsTer61 chr 15-78305554-TTC-T Novel	MutationTaster: Disease causing Prob = 1 PolyPhen-2: N/A SIFT: Damaging; score = 0.858 CADD: N/A DANN: N/A Varsome: N/A
2	2 (Male)	Hereditary gingival fibromatosis	Gingival fibromatosis	*TBC1D2B* variant NM_144572.2; NP_653173.1 c.2471A>G p.Tyr824Cys chr 15-78295750-T-C rs199928887 MAF = 0.00001301 *KREMEN2* variant NM_172229.3; NP_757384.1 c.892C>T p.Arg298Cys chr 16-3017162-C-T Novel	MutationTaster: Disease causing Prob = 0.999153978449563 PolyPhen-2: Probably damaging; score = 0.999 SIFT: Damaging; score = 0.01 CADD score = 22.8 DANN score = 0.9935 Varsome: Uncertain Significance MutationTaster: Disease causing Prob = 0.876965507429932 PolyPhen-2: Probably damaging; score = 0.994 SIFT: Tolerated; score = 0.18 CADD score = 23.5 ^1^ DANN score = 0.9981 ^2^ Varsome: Uncertain significance
	3 (Male)	Hereditary gingival fibromatosis	Gingival fibromatosis, epithelial dysplasia, dense mandibular bone		
	4 (Male)	Hereditary gingival fibromatosis	Gingival fibromatosis, dense mandibular bone		

^1^ The Combined Annotation Dependent Depletion (CADD); ≤22 means moderate benign, ≤23.2 means supporting benign, ≥25.6 means supporting pathogenic (https://varsome.com/about/resources/germline-implementation/, accessed on 14 April 2024). ^2^ Deleterious Annotation of genetic variants using Neural Networks (DANN); ≤0.915 means moderate benign, ≤0.974 means supporting benign, ≥0.999 means supporting pathogenic (https://varsome.com/about/resources/germline-implementation/, accessed on 14 April 2024).

## Data Availability

Data are contained within the article.

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
