# Peer review of "Genetic Variants in the TBC1D2B Gene Are Associated with Ramon Syndrome and Hereditary Gingival Fibromatosis"

_ijms, 2024, doi:10.3390/ijms25168867_

Round 1

Reviewer 1 Report

Comments and Suggestions for Authors

This study is significant as it introduces the role and importance of TBC1D2B in Ramon syndrome, especially in the pathogenesis of HGF with Ramon syndrome, which is very interesting. We look forward to further research. Thank you for your hard work, and I hope you will consider some of the contents.

1. Please use MeSH Terms for keywords. following: Ramon Syndrome Fibromatosis, Gingival Intellectual Disability Cherubism Hypertrichosis Arthritis, Juvenile Cambodia Exome Sequencing Tanzania Dysfunctional Brain Mutant Proteins Fibroma

2. --- was the cause of Ramon syndrome [6].  -> [6] letters more smaller than the others. Please correct it.

3. In the discussion section, the limitations of this study are outlined, along with suggestions for future research directions.

Comments on the Quality of English Language

The remarks concerning the quality of the English language are well-founded; it is a language that is relatively straightforward to comprehend.

Reviewer 2 Report

Comments and Suggestions for Authors

Comments on the Quality of English Language
